# Impact of herpes zoster vaccination on incident dementia: A retrospective study in two patient cohorts

**Jeffrey F. Scherrer** [1,2,3☯] *, **Joanne Salas** [1,2,3☯], **Timothy L. Wiemken** [3,4,5,6☯], **Daniel F. Hoft** [5,6,7☯], **Christine Jacobs** [1,3☯], **John E. Morley** [8☯]

**1** Department of Family and Community Medicine, Saint Louis University School of Medicine, St. Louis, MO, United States of America, **2** Harry S. Truman Veterans Administration Medical Center, Columbia, MO, United States of America, **3** The AHEAD Institute, Saint Louis University School of Medicine, St. Louis, MO, United States of America, **4** Department of Health and Clinical Outcomes Research, School of Medicine, Saint Louis University, Saint Louis, MO, United States of America, **5** Division of Infectious Diseases, Allergy, and Immunology, Department of Internal Medicine, Department of Medicine, School of Medicine, Saint Louis University, Saint Louis, MO, United States of America, **6** Saint Louis University Systems Infection Prevention Center, Center for Specialized Medicine, St. Louis, MO, United States of America, **7** Department of Molecular Microbiology & Immunology, Saint Louis University, Saint Louis, MO, United States of America, **8** Division of Geriatric Medicine, Saint Louis University School of Medicine, St. Louis, MO, United States of America

☯ These authors contributed equally to this work.

\* jeffrey.scherrer@health.slu.edu

**Data Availability Statement:** Data Availability Statement: Veterans Health Administration (VHA) may not be shared without interested persons obtaining VHA IRB and Data Access Request

## Abstract

### Background

Herpes zoster (HZ) infection increases dementia risk, but it is not known if herpes zoster vaccination is associated with lower risk for dementia. We determined if HZ vaccination, compared to no HZ vaccination, is associated with lower risk for incident dementia.

### Methods and findings

Data was obtained from Veterans Health Affairs (VHA) medical records (10/1/2008–9/30/2019) with replication in MarketScan® commercial and Medicare claims (1/1/2009-12/31/2018). Eligible patients were ≥65 years of age and free of dementia for two years prior to baseline (VHA n = 136,016; MarketScan n = 172,790). Two index periods (either start of 2011 or 2012) were defined, where patients either had or did not have a HZ vaccination. Confounding was controlled with propensity scores and inverse probability of treatment weighting. Competing risk (VHA) and Cox proportional hazard (MarketScan) models estimated the association between HZ vaccination and incident dementia in all patients and in age (65–69, 70–74, ≥75) and race (White, Black, Other) sub-groups. Sensitivity analysis measured the association between HZ vaccination and incident Alzheimer's dementia (AD). HZ vaccination at index versus no HZ vaccination throughout follow-up. VHA patients mean age was 75.7 (SD±7.4) years, 4.0% were female, 91.2% white and 20.2% had HZ vaccination. MarketScan patients mean age was 69.9 (SD±5.7) years, 65.0% were female and 14.2% had HZ vaccination. In both cohorts, HZ vaccination compared with no vaccination, was significantly associated with lower dementia risk (VHA HR = 0.69; 95%CI: 0.67–0.72;

approvals. MarketScan data is proprietary and the authors are prevented from sharing data per data use agreement. The authors do not have rights to share these data bases. While there is no fee involved in use of VHA data, investigators must have an appointment at their local VHA Research Service. Interested parties who obtain an affiliation will then be required to obtain IRB approval from their VHA site and complete data access request forms and finally sign a Data Use Agreement. To begin the process of obtaining VHA data, interested parties should study data access information at https://www.virec.research.va.gov/ MarketScan provides data bases containing medical claims for a fee. Special permissions are required after a fee is paid and these are contained in MarketScan Data Use Agreements. The cost of data may vary by the number of years of observation and sample size. Persons interested in purchasing MarketScan data should identify the data sources and contacts available at the following website: https://www.ibm.com/products/marketscan-research-databases/databases.

**Funding:** Benter Foundation award 2020-01 to JFS https://benterfoundation.org/ The sponsor played no role in study design, data collection, analysis, decision to publish or preparation of the manuscript.

**Competing interests:** The authors have declared that no competing interests exist.

MarketScan HR = 0.65; 95%CI:0.57–0.74). HZ vaccination was not related to dementia risk in MarketScan patients aged 65–69 years. No difference in HZ vaccination to dementia effects were found by race. HZ vaccination was associated with lower risk for AD.

## Conclusions

HZ vaccination is associated with reduced risk of dementia. Vaccination may provide non-specific neuroprotection by training the immune system to limit damaging inflammation, or specific neuroprotection that prevents viral cytopathic effects.

## Introduction

Worldwide, ten million new cases of dementia occur each year (1) and this number is expected to grow with an aging population [1]. Unfortunately, there are no interventions that prevent dementia.

Chronic bacterial, fungal, and viral infections have been identified as potential modifiable causes of Alzheimer's dementia (AD) [2]. An emerging "innate immune system dysregulation hypothesis" [3] links chronic infections to development of AD [4]. Inflammation and oxidation in chronic and/or recurrent infections in combination with other factors may exacerbate risk for AD and other dementias [5]. Numerous bacterial and viral infections have been associated with incident dementia and or AD [2, 3].

Herpes simplex virus has been associated with increased risk for dementias [5–7]. In addition, genetic risk markers for Alzheimer's Disease (AD) have been shown to interact with herpes simplex virus to increase likelihood of developing AD [8]. In large health insurance databases and patient registries, herpes zoster (HZ) infection also has been associated with an increased risk for dementia [9, 10]. Patients with vs. without herpes zoster ophthalmicus had nearly a 3-fold increased risk for developing dementia [11]. Furthermore, patients with a history of herpes simplex or HZ infection and received antivirals, have a lower risk for dementia than patients with a history of these infections who did not receive antivirals [6, 9, 10].

Self-reported receipt of diphtheria/tetanus and polio vaccinations are associated with a 60% and 46% lower risk for dementia, respectively [12]. Cohort studies of patients with chronic kidney disease and chronic obstructive pulmonary disease found a dose-dependent protective effect of influenza immunization on dementia [13, 14]. However, there are several weaknesses in this literature including reliance on self-reported vaccination history [12] and electronic health record based cohorts limited to persons with either kidney disease or chronic obstructive pulmonary disease [13, 14]. To our knowledge there are no studies of the association between HZ vaccination and incident dementia.

The purpose of this study was to determine if HZ vaccination is associated with incident dementia. Using a cohort of Veterans Health Affairs (VHA) patient data we determined 1) if patients who receive an HZ vaccination, compared with those who do not, have a lower risk for incident dementia, 2) if the association between HZ vaccination and incident dementia differs by age groups (65–69, 70–74, $\geq$ 75 years of age) and race (White, Black, Other), 3) if the association between HZ vaccination and incident dementia is explained by HZ infection during follow-up, 4) if HZ vaccination is associated with lower risk for AD and 5) if results from the VHA cohort can be replicated in a private sector, medical claims database.

## Methods

Veterans Health Administration (VHA) administrative medical record data for fiscal years (FY) FY09 –FY19 (10/1/2008–9/30/2019) were used to create variables. VHA data includes ICD-9-CM and ICD-10-CM diagnoses, vaccinations, prescription fills, laboratory results, vital signs, and demographic data. VHA linked Medicare claims and Part-D pharmacy claims were used to obtain diagnoses codes, laboratory values, and prescription fills.

IBM® MarketScan® Commercial Claims and Medicare Supplemental databases from 1/1/2009-12/31/2018 (calendar year (CY)2009-CY2018) were used to determine if results from VHA patients could be replicated in private sector medical claims data. The IBM® Market-Scan® data contained individual-level, de-identified, healthcare claims information from inpatient and outpatient encounters, academic and non-academic health centers and private and government health insurance.

### Eligibility

Detailed variable definitions are shown in S1 Table in S1 Appendix. Patients with more well-visits are more likely to receive adult vaccinations [15]. Therefore sampling began by selecting patients with at least three well visits during the entire observation period. Patients were 50 years or older at first well visit (VHA– 458,460; MarketScan– 4,989,703). Two index dates were defined for both cohorts; the start of CY2011 and 2012 for MarketScan (1/1/2011 and 1/1/2012) and FY2011 and 2012 for VHA (10/1/2010 and 10/1/2011). Thus, in VHA data there was 8–9 years follow-up and in MarketScan 7–8 years follow-up available. In both cohorts, patients were eligible if they were free of dementia and free of conditions associated with cognitive decline (e.g., Creutzfeldt-Jakob disease) for two years prior to index date. We selected patients ≥ 65 years old who had > 90 days of follow-up time after index. Patients without HZ vaccination at index were free of HZ vaccination in follow-up. Patients not meeting eligibility criteria in 2011 were used to sample for the 2012 index date.

After applying eligibility criteria and removing patients with missing demographic data (2595 missing marital status, 74 missing race, 1 missing gender), the eligible VHA cohort contained 136,016 patients. The eligible MarketScan cohort contained 172,790 patients. Cohort creation is illustrated in S1a and S1b Fig in S1 Appendix.

**Exposure.** Receipt of a HZ vaccination was measured by CPT codes: 90710, 90716, 90736, 90750, and product names: Proquad, Varivax, Zostavax, and Shingrix. HZ vaccination status was defined at index date; patients with HZ vaccination must have had at least one CPT code or drug fill by index date. There were 24,612 MarketScan and 27,419 VHA patients with HZ vaccination by baseline. Proquad (n = 7 VHA, n = 3 MarketScan), Varivax (n = 770 VHA, n = 79 MarketScan), and Zostavax (n = 26,642 VHA, n = 24,530 MarketScan). All HZ vaccine products were measured during follow-up to exclude unvaccinated patients who may have received a vaccine in follow-up. Follow-up time was defined as months from index date to dementia or censoring. MarketScan patients were censored at last available inpatient or outpatient claim. VHA patients were censored at last available inpatient, outpatient, or Medicare claim, or death. MarketScan data did not include mortality data.

**Outcome.** Incident dementia was defined by diagnostic codes on two separate days in any 12-month period. The first of the two codes for dementia was the date of onset (S1 Table in S1 Appendix). We have used this dementia diagnostic algorithm in studies of metformin and incident dementia [16] and we found good agreement between the diagnostic algorithm and the Mini Mental State Exam and the Saint Louis University Mental Status Examination scores indicating dementia [17].

**Covariates.** Unless stated, the same covariates were used in analyses for both the VHA and MarketScan cohorts. Covariates included demographics, geographic region, health care utilization, comorbid conditions and prescription medications. All covariates were measured from the start of the observation period up to index date.

Demographic variables were age, race (VHA only), gender, and marital status (VHA only). We adjusted for United States geographic regions (i.e. Northeast, North Central, South, West–see S1 Table for definitions in S1 Appendix) to control for variation in clinical practice and vaccination rates across regions. To control for detection bias, we computed the average number of outpatient healthcare encounters/medical claims per month and used the distribution to classify patients into high health care users (top 25th percentile) vs. not-high utilizers. We controlled for the number of well visits prior to baseline. In VHA data, health insurance was defined as having only VHA insurance compared to VHA insurance plus other health insurance.

Several comorbid conditions are associated with dementia [18] and obtaining vaccinations [19]. Using ICD-9-CM and ICD-10-CM codes, we controlled for type 2 diabetes, obesity, hypertension, stroke, ischemic heart disease, congestive heart failure, atrial fibrillation, asthma, chronic obstructive pulmonary disease, traumatic brain injury, Vitamin B12 deficiency, depression, anxiety disorder, nicotine dependence, and alcohol and drug abuse/dependence. BMI and social history health factor information were also available in the VHA medical record for defining obesity and smoking. Medications that could alter the risk of dementia were included as covariates. We controlled for sustained (defined as receipt of two or more prescriptions in any six-month period) use of anticholinergics, non-steroidal anti-inflammatory drugs (NSAIDS), anti-hypertensives, statins, steroids, antiviral medications, metformin, and sulfonylurea.

**Control for confounding.** Because receiving HZ vaccination is not random, we used propensity scores (PS) and inverse probability of treatment weighting (IPTW) to balance covariates between patients receiving and not receiving HZ vaccination (details in S1 Appendix).

**Primary analysis.** All analyses were performed with SAS v9.4 (SAS Institute, Cary, NC) at a two-tailed alpha = 0.05. Unweighted bivariate analyses used chi-square tests and SMD% for effect size to estimate differences between HZ and no-HZ patients at baseline. In VHA data, competing-risk survival regression models [20] were used to control for bias in estimated endpoint probabilities due to the existence of a competing event (mortality) precluding detection of dementia. Mortality data was not available in MarketScan data. In MarketScan, Cox proportional hazard models were used. In both cohorts, models before and after IPTW were calculated and measures of association expressed as hazard ratios and 95% confidence intervals, overall and stratified by age and race (only VHA). Prior studies have shown black persons have a higher incidence of dementia [21]. thus, HZ-dementia hazard ratios were compared between White, Black, and Other race. Effect modification by age or race was assessed using an interaction term of age group (or race) and HZ vaccination in overall models. Weighted models used robust, sandwich-type variance estimators to calculate confidence intervals and p-values [22]. Examination of a time-dependent interaction term of HZ vaccination and log (follow-up time) in all models indicated that the proportional hazard assumption was met ($p > 0.05$) for all models.

**Secondary analysis.** We modeled HZ infection and any antiviral therapy that occurred after index and prior to the end of follow-up in final weighted models. Any attenuation of the HZ vaccination effect may indicate that HZ vaccination has a possible indirect effect on dementia through HZ infection or antiviral therapy [23].

We computed the false discovery rate (FDR) adjusted p-value (q-value) to adjust for multiple comparisons and test for significance [24].

Because mechanisms leading to AD may differ from those leading to other forms of dementia, we conducted sensitivity analysis to measure the association between HZ vaccination and incident AD in overall samples.

To generate the most precise estimate of incident dementia risk associated with HZ vaccination, overall weighted model results from VHA and MarketScan were combined using an inverse variance weighted fixed effects meta-analysis [25].

The e-values for both the hazard ratio and upper confidence limit were calculated to determine if unmeasured confounding may explain our results [26]. The e-value for the point estimate is the minimum strength of association that is needed for an unmeasured confounder to have with both the exposure and outcome to completely explain the association of HZ vaccination and dementia. The e-value for the upper confidence interval is the minimum strength of association an unmeasured confounder would need for both the exposure and outcome to completely explain away the upper confidence interval (i.e. to include the null). To investigate the potential effect of selection bias, a bias analysis using an upper bounding factor approach was conducted [27]. Like e-values, this approach determines the strength of association a selection factor would have to have with the exposure and outcome to produce a significant finding when the true association is null. The e-values for unmeasured confounding and selection bias were computed for the meta-analysis hazard ratio.

*IRB review*. The Saint Louis University IRB approved the research as a non-human subjects study because this was a retrospective cohort study of anonymized medical record and claims data. Investigators only had access to anonymized data.

## Results

VHA patients averaged 75.7 (SD±7.4) years of age, 4.0% were female and 91.2% were white. MarketScan patients averaged 69.9 (SD±5.7) years of age and 65.0% were female. The prevalence of comorbid conditions and prescription medications were markedly higher in the VHA patient sample (Table 1). The distribution of post-vaccination HZ infection and HZ antiviral use is reported in the S1 Appendix.

As shown in Table 2, 20,2% of VHA patients and 14.2% of the MarketScan sample received a HZ vaccination. Patients 70–74 years old were more prevalent among VHA patients who were vaccinated compared to not vaccinated VHA patients (SMD% = 12.4) and in both cohorts, those ≥75 years of age were less prevalent among those vaccinated than not vaccinated (SMD% = -11.4). Among VHA patients, white race (SMD% = 27.5) and being married were positively associated with HZ vaccination (SMD% = 21.6), while black race was inversely associated with vaccination (SMD% = -28.7).

In VHA and MarketScan data, geographic region was associated with HZ vaccination. Having only VHA insurance was inversely related to HZ vaccination (SMD% = -24.2). High health care use in the VHA, but not in MarketScan, was less prevalent among patients with HZ vaccination (SMD% = -12.7). In MarketScan, but not in VHA, patients with no well visits were less prevalent (SMD% = -22.8) among those vaccinated while those with three or more well visits were more prevalent among patients vaccinated (SMD% = 32.2). Among VHA patients, type 2 diabetes, congestive heart failure, and nicotine dependence were more common among those without an HZ vaccination (SMD%>10). Comorbidities in the MarketScan sample were not associated with HZ vaccination. In both patient samples, statin medication use was more common among those with an HZ vaccination.

As shown in the S2 Table in S1 Appendix, IPTW balanced (SMD% <10%) all baseline covariates between patients with and without HZ vaccination. Variables were also balanced in each age and race subgroup (SMD%<10%, results not shown). Among VHA patients,

**Table 1. Characteristics (n, (%)) of ≥ 65 years old patients, VHA (n = 136,016) and MarketScan (n = 172,790).**

| Covariates | VHA | MarketScan |
|---|---|---|
| | **(n = 136,016)** | **(n = 172,790)** |
| Index year | | |
| 2011 | 116678 (85.8) | 115968 (67.1) |
| 2012 | 19338 (14.2) | 56822 (32.9) |
| *Sociodemographic-related* | | |
| Age, mean (±sd) | 75.7 (±7.4) | 69.9 (±5.7) |
| Age category | | |
| 65–69 | 36837 (27.1) | 105274 (60.9) |
| 70–74 | 24516 (18.0) | 33814 (19.6) |
| ≥ 75 | 74663 (54.9) | 33702 (19.5) |
| Female gender | 5466 (4.0) | 112287 (65.0) |
| Race | | |
| White | 124118 (91.2) | - |
| Black | 10421 (7.7) | - |
| Other | 1477 (1.1) | - |
| Married | 95175 (70.0) | - |
| Region | | |
| Northeast | 16484 (12.1) | 64170 (37.1) |
| North central | 57381 (42.2) | 32963 (19.1) |
| South | 43995 (32.4) | 42321 (24.5) |
| West | 14943 (1.0) | 31145 (18.0) |
| Unknown | 3213 (2.4) | 2191 (1.3) |
| VHA only insurance | 24938 (18.3) | - |
| High healthcare utilization | 34595 (25.4) | 43190 (25.0) |
| # well visits, mean (±sd) | 1.1 (±2.5) | 1.6 (±1.0) |
| # well visits, category | | |
| 0 | 80771 (59.4) | 22787 (13.2) |
| 1–2 | 36627 (26.9) | 125994 (72.9) |
| ≥ 3 | 18618 (13.7) | 24009 (13.9) |
| *Comorbidities* | | |
| Type II Diabetes | 44063 (32.4) | 25228 (14.6) |
| Obesity | 46186 (34.0) | 9099 (5.3) |
| Hypertension | 112597 (82.8) | 102720 (59.5) |
| Stroke | 6450 (4.7) | 2973 (1.7) |
| Ischemic heart disease | 57028 (41.9) | 24865 (14.4) |
| Congestive heart failure | 19858 (14.6) | 5054 (2.9) |
| Atrial fibrillation | 22170 (16.3) | 9392 (5.4) |
| Asthma | 10165 (7.5) | 12556 (7.3) |
| COPD | 28189 (20.7) | 10056 (5.8) |
| Traumatic brain injury | 3600 (2.7) | 2856 (1.7) |
| Vitamin B12 deficiency | 7666 (5.6) | 4244 (2.5) |
| Depression | 12177 (9.0) | 6196 (3.6) |
| Anxiety disorder [a] | 10655 (7.8) | 4964 (2.9) |
| Nicotine dependence | 32083 (23.6) | 7612 (4.4) |
| Alcohol abuse/dependence | 5414 (4.0) | 710 (0.4) |
| Drug abuse/dependence | 1512 (1.1) | 236 (0.1) |
| *Medications [b]* | | |

*(Continued)*

**Table 1.** (Continued)

| Covariates | VHA | MarketScan |
|---|---|---|
| | **(n = 136,016)** | **(n = 172,790)** |
| Anticholinergics | 17113 (12.6) | 19073 (11.0) |
| NSAIDS | 17831 (13.1) | 15423 (8.9) |
| Antihypertensives | 98013 (72.1) | 82631 (47.8) |
| Statins | 79708 (58.6) | 63924 (3.0) |
| Steroids | 14337 (10.5) | 9297 (5.4) |
| Antivirals | 1684 (1.2) | 3321 (1.9) |
| Metformin | 17640 (13.0) | 11476 (6.6) |
| Sulfonylurea | 16754 (12.3) | 5685 (3.3) |

[a]Anxiety disorders = panic disorder, OCD, social phobia, GAD, Anxiety NOS.

[b]Medications = sustained use prior to index (at least 2 fills in a 6-month period).

stabilized weights ranged from 0.22–9.96 with a mean = 0.99 (SD = 0.48); and in MarketScan, stabilized weights ranged from 0.16–9.93 with a mean = 1.00 (SD = 0.29).

The overall median duration of follow-up was 95 (IQR = 57–106) months in VHA and 37 (IQR = 24–60) months in MarketScan. Median follow-up time among those with HZ vaccine was 96 (IQR = 87–107) months in VHA and 38 (IQR = 24–63) months in MarketScan. Median follow-up time for those without HZ vaccine was 93 (IQR = 52–106) months in VHA and 37 (IQR = 24–60) months in MarketScan. There were 22,710 new dementia cases in the VHA and 3,230 in MarketScan (cumulative incidence: 16.7% and 1.9%, respectively). Among those who developed dementia, the median follow-up time, in months, to incident dementia was: a) 55 (IQR = 30–75) in VHA and 40 (IQR = 25–60) in MarketScan for those without HZ vaccination; and b) 64 (IQR = 43–79) in VHA and 46 (IQR = 27–66) in MarketScan for those with HZ vaccination."

The unadjusted incidence rate of dementia for HZ vaccination compared to no vaccination in VHA was 148.2/10,000PY and 283.7/10,000PY, respectively, and for MarketScan, 36.5/10,000PY and 52.8/10,000PY, respectively, (See Table 3).

Results of competing risk survival models estimating the association between HZ vaccination and incident dementia in VHA are shown in Fig 1 and S3 Table in S1 Appendix. After controlling for confounding in weighted data, VHA patients with, compared to those without, an HZ vaccination were significantly less likely to develop dementia (HR = 0.69; 95%CI:0·67–0.72). This association remained after including HZ infection and antiviral treatment between baseline and end of follow-up (S3 Table in S1 Appendix). As shown in Fig 1 and S4 Table in S1 Appendix, HZ vaccination, compared to no vaccination in MarketScan patients was significantly associated with lower dementia risk (HR = 0.65; 95%CI:0.57–0.74). This association remained after including post-index HZ infection and antiviral therapy in survival models.

The association between HZ vaccination and incident dementia differed by age group in both VHA and MarketScan (interaction term q = 0.0002 and q = 0.008, respectively). The association between HZ vaccination and dementia was weakest among VHA patients age 70–74 (HR = 0.83, 95% CI: 0.76–0.91) but relatively similar among patients age 65–69 (HR = 0.61, 95% CI: 0.54–0.68) and ≥75 (HR = 0.66, 95% CI: 0.63–0.68). In MarketScan patients, there was no association between vaccination status and incident dementia among patients 65 to 69 years of age. The association between HZ vaccination and incident dementia was similar among the other age groups in MarketScan data (Fig 1). In VHA data, the effect of HZ vaccination on incident dementia did not differ by race group (q = 0.436).

**Table 2. Characteristics (%) of ≥ 65 years old patients by herpes zoster (HZ) vaccination status, VHA (n = 136,016) and MarketScan (172,790).**

| Covariates | VHA | | | | MarketScan | | | |
|---|---|---|---|---|---|---|---|---|
| | No HZ vaccination (n = 108,597) | HZ vaccination (n = 27,419) | Unwtd p-value | Unwtd. SMD% | No HZ vaccination (n = 148,178) | HZ vaccination (n = 24,612) | Unwtd p-value | Unwtd. SMD% |
| Index fiscal year | | | < .001 | | | | < .001 | |
| 2011 | 97516 (89.8) | 19162 (69.9) | | -5.2 | 103073 (69.6) | 12895 (52.4) | | -35.8 |
| 2012 | 11081 (10.2) | 8257 (30.1) | | 51.2 | 45105 (30.4) | 11717 (47.6) | | 35.8 |
| *Sociodemographic-related* | | | | | | | | |
| Age category | | | < .001 | | | | < .001 | |
| 65–69 | 29243 (26.9) | 7594 (27.7) | | 1.7 | 89461 (60.4) | 15813 (64.2) | | 8.0 |
| 70–74 | 18502 (17.0) | 6014 (21.9) | | 12.4 | 28896 (19.5) | 4918 (20.0) | | 1.2 |
| ≥ 75 | 60852 (56.0) | 13811 (50.4) | | -11.4 | 29821 (20.1) | 3881 (15.8) | | -11.4 |
| Female gender | 4333 (4.0) | 1133 (4.1) | .28 | 0.7 | 96845 (65.4) | 15442 (62.7) | < .001 | -5.5 |
| Race | | | < .001 | | | | | |
| White | 97604 (89.9) | 26514 (96.7) | | 27.5 | | | | |
| Black | 9763 (9.0) | 658 (2.4) | | -28.7 | - | - | | |
| Other | 1230 (1.1) | 247 (0.9) | | -2.3 | - | - | | |
| Married | 73895 (68.1) | 21280 (77.6) | < .001 | 21.6 | - | - | | |
| Region | | | < .001 | | | | < .001 | |
| Northeast | 13191 (12.2) | 3293 (12.0) | | -0.4 | 57385 (38.7) | 6785 (27.6) | | -23.9 |
| North central | 40410 (37.2) | 16971 (61.9) | | 50.9 | 27071 (18.3) | 5892 (23.9) | | 13.9 |
| South | 39826 (36.7) | 4169 (15.2) | | -50.5 | 37208 (25.1) | 5113 (20.8) | | -10.3 |
| West | 11961 (11.0) | 2982 (10.9) | | -0.4 | 24732 (16.7) | 6413 (26.1) | | 23.0 |
| Unknown | 3209 (3.0) | < 5 | | -24.5 | 1782 (1.2) | 409 (1.7) | | 3.9 |
| VHA only insurance | 21822 (20.1) | 3116 (11.4) | < .001 | -24.2 | - | - | - | - |
| High healthcare utilization | 28800 (26.5) | 5795 (21.1) | < .001 | -12.7 | 37104 (25.0) | 6086 (24.7) | .30 | -0.7 |
| # well visits, category | | | < .001 | | | | < .001 | |
| 0 | 6572 (59.9) | 15699 (57.3) | | -5.4 | 21021 (14.2) | 1766 (7.2) | | -22.8 |
| 1–2 | 28958 (26.7) | 7669 (28.0) | | 2.9 | 109159 (73.7) | 16835 (68.4) | | -11.6 |
| ≥ 3 | 14567 (13.4) | 4051 (14.8) | | 3.9 | 17998 (12.2) | 6011 (24.4) | | 32.2 |
| *Comorbidities* | | | | | | | | |
| Type II Diabetes | 36358 (33.5) | 7705 (28.1) | < .001 | -11.7 | 21983 (14.8) | 3245 (13.2) | < .001 | -4.8 |
| Obesity | 35734 (32.9) | 10452 (38.1) | < .001 | 10.9 | 7635 (5.2) | 1464 (6.0) | < .001 | 3.5 |
| Hypertension | 90415 (83.3) | 22182 (80.9) | < .001 | -6.2 | 88640 (59.8) | 14080 (57.2) | < .001 | -5.3 |
| Stroke | 5520 (5.1) | 930 (3.4) | < .001 | -8.4 | 2555 (1.7) | 418 (1.7) | .77 | -0.2 |
| Ischemic heart disease | 46555 (42.9) | 10473 (38.2) | < .001 | -9.5 | 21652 (14.6) | 3213 (13.1) | < .001 | -4.5 |
| Congestive heart failure | 17185 (15.8) | 2673 (9.8) | < .001 | -18.3 | 4459 (3.0) | 595 (2.4) | < .001 | -3.6 |
| Atrial fibrillation | 18330 (16.9) | 3840 (14.0) | < .001 | -8.0 | 8133 (5.5) | 1259 (5.1) | .02 | -1.7 |
| Asthma | 8151 (7.5) | 2014 (7.4) | .37 | -0.6 | 10621 (7.2) | 1935 (7.9) | .001 | 2.6 |
| COPD | 23914 (22.0) | 4275 (15.6) | < .001 | -16.5 | 8792 (5.9) | 1264 (5.1) | < .001 | -3.5 |
| Traumatic brain injury | 3058 (2.8) | 542 (2.0) | < .001 | -5.5 | 2377 (1.6) | 479 (2.0) | < .001 | 2.6 |
| Vitamin B12 deficiency | 6451 (5.9) | 1215 (4.4) | < .001 | -6.8 | 3644 (2.5) | 600 (2.4) | .84 | -0.1 |
| Depression | 9923 (9·1) | 2254 (8.2) | < .001 | -3.3 | 5108 (3.5) | 1088 (4.4) | < .001 | 5.0 |
| Anxiety disorder [a] | 8705 (8.0) | 1950 (7.1) | < .001 | -3.4 | 4152 (2.8) | 812 (3.3) | < .001 | 2.9 |
| Nicotine dependence | 26612 (24.5) | 5471 (20.0) | < .001 | -11.0 | 6464 (4.4) | 1148 (4.7) | .03 | 1.5 |

*(Continued)*

**Table 2.** (Continued)

| Covariates | VHA | | | | MarketScan | | | |
|---|---|---|---|---|---|---|---|---|
| | No HZ vaccination (n = 108,597) | HZ vaccination (n = 27,419) | Unwtd p-value | Unwtd. SMD% | No HZ vaccination (n = 148,178) | HZ vaccination (n = 24,612) | Unwtd p-value | Unwtd. SMD% |
| Alcohol abuse/ dependence | 4533 (4.2) | 881 (3.2) | < .001 | -5.1 | 580 (0.4) | 130 (0.5) | .002 | 2.0 |
| Drug abuse/ dependence | 1353 (1.3) | 159 (0.6) | < .001 | -7.0 | 189 (0.1) | 47 (0.2) | .01 | 1.6 |
| *Medications* [b] | | | | | | | | |
| Anticholinergics | 13934 (12.8) | 3179 (11.6) | < .001 | -3.9 | 15796 (10.7) | 3277 (13.3) | < .001 | 8.2 |
| NSAIDS | 14047 (12.9) | 3784 (13.8) | < .001 | 2.5 | 12667 (8.6) | 2756 (11.2) | < .001 | 8.9 |
| Antihypertensives | 77829 (71.7) | 20184 (73.6) | < .001 | 4.4 | 70151 (47.3) | 12480 (50.7) | < .001 | 6.7 |
| Statins | 61788 (56.9) | 17920 (65.4) | < .001 | 17.4 | 53382 (36.0) | 10542 (42.8) | < .001 | 14.0 |
| Steroids | 11643 (10.7) | 2694 (9.8) | < .001 | -3.0 | 7813 (5.3) | 1484 (6.0) | < .001 | 3.3 |
| Antivirals | 1324 (1.2) | 360 (1.3) | .21 | 0.8 | 2660 (1.8) | 661 (2.7) | < .001 | 6.0 |
| Metformin | 14003 (12.9) | 3637 (13.3) | .10 | 1.1 | 9818 (6.6) | 1658 (6.7) | .52 | 0.4 |
| Sulfonylurea | 13899 (12.8) | 2855 (10.4) | < .0001 | -7.5 | 4964 (3.4) | 721 (2·9) | .001 | -2.4 |

[a]Anxiety disorders = panic disorder, OCD, social phobia, GAD, Anxiety NOS
[b]Medications = sustained use prior to index (at least 2 fills in a 6-month period).

S5 Table in S1 Appendix shows crude, descriptive AD incidence, and the associations between HZ vaccination and AD in both VHA and MarketScan are shown in S6 Table in S1 Appendix. Overall incidence rate of AD was 80.0/10,000PY in VHA and 18.0/10,000PY in MarketScan. Weighted survival models showed that HZ vaccination was associated with a 25% decreased risk of AD in VHA (HR = 0.75, 95% CI: 0.71–0.80) and 30% decreased risk in MarketScan (HR = 0.70, 95% CI: 0.55–0.88).

Meta-analysis revealed a 31% decreased risk of dementia associated with HZ vaccination (HR = 0.69, 95% CI: 0.66–0.71). The e-value for unmeasured confounding was 2.26 for the point estimate and 2.17 for the confidence interval. The selection bias e-value was 1.70 for the point estimate and 1.66 for the confidence interval.

## Discussion

Our results indicate adult HZ vaccination is associated with a 31% reduced risk for dementia. This association is independent of numerous sociodemographic factors, comorbid conditions, and medication use. HZ infection and antiviral therapy after HZ vaccination did not alter results. Thus, the association between HZ vaccination and incident dementia may not be explained by lower infection risk or antiviral therapy. Lower risk for dementia following HZ vaccination did not differ by race. Our sensitivity analyses indicates that the reduced risk for dementia following HZ vaccination also applies to AD. While the magnitude of effect was marginally less for risk of AD, our results support an association between HZ vaccination and reduced risk for dementias, including AD. Vaccinations may reduce risk for other forms of dementia, such as vascular dementia, by lowering inflammation which is common in cardiovascular conditions. We replicated findings in two patient populations 65 years of age and older. Obtaining very similar results in these patient groups that differ in demographic and clinical characteristics strengthens our conclusions.

Our results are consistent with a previous study which found that subjects who self-reported a past diphtheria/tetanus, influenza, and polio vaccination had a 60%, 25% and 40% lower risk for Alzheimer's Disease compared to subjects who reported receiving none of these

**Table 3. VHA and MarketScan–association between herpes zoster (HZ) vaccination and dementia in follow-up, cumulative incidence % and incidence rate per 10,000 person-years (PY), patients ≥ 65 years old.**

| Age group | VHA | | | | MarketScan | | | |
|---|---|---|---|---|---|---|---|---|
| | Total n | Dementia events | Cumulative incidence % | Incidence rate per 10,000PY | Total n | Dementia events | Cumulative incidence % | Incidence rate per 10,000PY |
| All ages | | | | | | | | |
| *Overall* | *136,016* | *22,710* | *16.7%* | *253.0/10,000PY* | *172,790* | *3,230* | *1.9%* | *50.0/10,000PY* |
| No vaccination | 108,597 | 19,678 | 18.1% | 283.7/10,000PY | 148,178 | 2,888 | 2.0% | 52.8/10,000PY |
| HZ vaccination | 27,419 | 3,032 | 11.1% | 148.2/10,000PY | 24,612 | 447 | 1.4% | 36.5/10,000PY |
| | | | p < .001 | p < .001 | | | p < .001 | p < .001 |
| Age 65–69 | | | | | | | | |
| *Overall* | *36,837* | *2,510* | *6.8%* | *92.0/10,000PY* | *105,274* | *566* | *0.5%* | *15.0/10,000PY* |
| No vaccination | 29,243 | 2,209 | 7.6% | 104.2/10,000PY | 89,461 | 482 | 0.5% | 15.0/10,000PY |
| HZ vaccination | 7,594 | 301 | 4.0% | 49.8/10,000PY | 15,813 | 84 | 0.5% | 14.4/10,000PY |
| | | | p < .001 | p < .001 | | | p = .904 | p = .712 |
| Age 70–74 | | | | | | | | |
| *Overall* | *24,516* | *2,816* | *11.5%* | *159.0/10,000PY* | *33,814* | *674* | *2.0%* | *50.0/10,000PY* |
| No vaccination | 18,502 | 2,385 | 12.9% | 183.5/10,000PY | 28,896 | 586 | 2.0% | 51.3/10,000PY |
| HZ vaccination | 6,014 | 431 | 7.2% | 91.0/10,000PY | 4,918 | 88 | 1.8% | 43.8/10,000PY |
| | | | p < .001 | p < .001 | | | p = .268 | p = .165 |
| Age ≥ 75 | | | | | | | | |
| *Overall* | *74,663* | *17,384* | *23.3%* | *388.0/10,000PY* | *33,702* | *1,990* | *5.9%* | *157.0/10,000PY* |
| No vaccination | 60,852 | 15,084 | 24.8% | 428.9/10,000PY | 29,821 | 1,820 | 6.1% | 162.8/10,000PY |
| HZ vaccination | 13,811 | 2,300 | 16.7% | 237.7/10,000PY | 3,881 | 170 | 4.4% | 112.1/10,000PY |
| | | | p < .001 | p < .001 | | | p < .001 | p < .001 |

Note: PY = person-years.

vaccinations [12]. Our findings are also consistent with two retrospective cohort studies of the Taiwan National Health Insurance Database that documented influenza vaccination was associated with a 35% lower risk among patients with chronic kidney disease [13], and 32% lower risk in patients with chronic obstructive pulmonary disease [14]. The hazard ratios obtained in our two patient samples are consistent with this prior work. However, we believe the current study has overcome limitations of prior work which included potential recall errors and reduced generalizability due to studies of cohorts with existing chronic conditions.

There are several potential explanations for our results. People who obtain vaccinations, compared to those who do not, have higher education, greater income, and social support [28–30]. We lacked measures of education, income, social support, and isolation; however, in the VHA cohort we controlled for access to non-VHA insurance which is a proxy for higher income. In the VHA sample we controlled for proxies of social support and isolation with our marital status variable. We are unable to exclude the possibility that HZ vaccination is in a causal pathway from higher education, income, social support to incident dementia.

HZ vaccination, compared to no HZ vaccination could have specific neuroprotective effects that reduce CNS inflammation and/or block viruses from invading the brain. Zoster disease

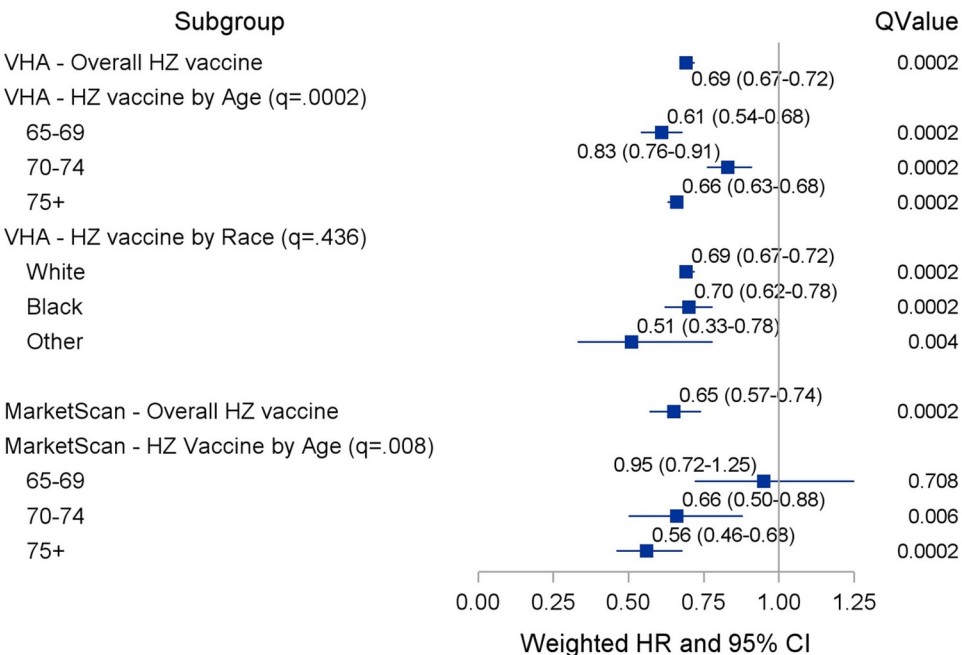

**Fig 1. Results from weighted competing risk (VHA) and Cox proportional hazard (MarketScan) models–association (HR (95% CI)) of HZ vaccination and incident dementia.**

represents clinically significant reactivation of a latent chronic viral infection. It is possible that vaccine-induced prevention of clinically detected and/or asymptomatic zoster reactivation events may limit associated CNS inflammation, and thereby also reduce the subsequent neuronal damage.

We observed that in VHA patients, 7.4% of patients without HZ vaccination and 6.9% of patients with HZ vaccine had at least one ICD diagnostic code for HZ infection in follow-up. This suggests a modest effect of these vaccinations on preventing HZ. This finding, that vaccination had little impact on HZ infection in our studied populations may be explained by the possibility that preventing HZ infection/reactivation is not the mechanism by which vaccination is associated with lower dementia risk.

HZ vaccination may have non-specific effects that are associated with decreased dementia by mitigating abnormal neuroinflammation mediated by cerebral innate immune cells which lead to autophagy, apoptosis, and neuronal cell damage [31]. Similarly, vaccination may reduce proinflammatory cytokines that can increase synthesis and inhibit degradation of amyloid-beta and increase phosphorylation of tau [32]. Lower mortality has been observed in children who received measles vaccination and among persons who received the bacillus Camette-Guérin (BCG) vaccination compared to those who did not [33]. This effect may be due to training the immune response leading to better resistance to a wide range of bacteria and viruses [33]. Similarly, trained immunity could maximize the capacity for appropriate immunity while minimizing the potential for inflammatory-driven onset of dementia. HZ vaccination alone, or paired with a lifetime of appropriate vaccination, may have non-specific effects that protect against inflammation leading to β amyloid build up and neurodegeneration. The similar magnitude of association between HZ vaccination and incident dementia in VHA and MarketScan cohorts paired with estimates from self-reported vaccination [12] and influenza vaccination of high risk patients [13, 14] lend credence that our observations reflect non-specific vaccination effects.

## Limitations

Our results should be interpreted in the context of study limitations. The observation time did not permit modeling Shingrix exposure which was released in the United States in late 2017. Our study was based on nationally distributed data in the United States and may not generalize to other regions of the world. Misclassification and unmeasured/residual confounding are inherent limitations in retrospective cohort studies. If we mistakenly classified cases of dementia as non-cases, this would most likely lead to conservative hazard ratios. It is possible that unmeasured confounding biased our analysis. The E-value of 2.26 indicates that an unmeasured confounder could explain our observations if it had a strong association, i.e. 2.26, with both HZ vaccination and dementia. We are unable to identify such a strong confounder. The selection bias e-value was 1.70. An unaccounted for selection factor would have to be associated with both HZ vaccination and incident dementia at a magnitude of 1.70 to produce the observed hazard ratio of 0.69 when the true hazard ratio was 1.0. We explored risks associated with dementia and HZ vaccination and none of our measured variables reached those thresholds for both HZ vaccination and dementia risk. We are also not aware of any unmeasured factors with this magnitude of association with HZ vaccination and incident dementia.

Receipt of other vaccinations may be associated with increased likelihood of receiving HZ vaccination and incident dementia. Our conclusions did not change in post-hoc analysis that controlled for receipt of influenza and pneumococcal vaccinations. While gender differences are an important component of the larger effort to determine if vaccination may reduce risk of dementia, an appropriately comprehensive evaluation of gender effects is beyond the objectives of this study.

## Conclusions

We observed HZ vaccination was associated with a 31% lower risk for dementia. Studies of other common adult vaccinations, prospective studies and clinical trials are needed to elucidate whether this is due to a specific or non-specific neuroprotective effect. Additional research may establish vaccinations as an inexpensive and accessible intervention to markedly reduce dementia risk.

## Supporting information

**S1 Appendix.**
(DOCX)

## Acknowledgments

The views expressed here do not necessarily reflect those of the Veterans Administration.

Prior presentations: none.

Dr. Scherrer and Ms. Joanne Salas had full access to all the data in the study and take responsibility for the integrity of the data and the accuracy of the data analysis.

## Author Contributions

**Conceptualization:** Jeffrey F. Scherrer, Timothy L. Wiemken, Daniel F. Hoft, Christine Jacobs, John E. Morley.

**Data curation:** Jeffrey F. Scherrer.

**Formal analysis:** Joanne Salas.

**Funding acquisition:** Jeffrey F. Scherrer, Timothy L. Wiemken.

**Investigation:** Jeffrey F. Scherrer, Timothy L. Wiemken, Daniel F. Hoft, John E. Morley.

**Methodology:** Jeffrey F. Scherrer, Joanne Salas, Timothy L. Wiemken, Christine Jacobs.

**Project administration:** Jeffrey F. Scherrer.

**Resources:** Jeffrey F. Scherrer.

**Software:** Joanne Salas.

**Supervision:** Jeffrey F. Scherrer, Timothy L. Wiemken.

**Validation:** Joanne Salas, Timothy L. Wiemken.

**Writing – original draft:** Jeffrey F. Scherrer, Joanne Salas, Timothy L. Wiemken, Daniel F. Hoft, Christine Jacobs, John E. Morley.

**Writing – review & editing:** Jeffrey F. Scherrer, Joanne Salas, Timothy L. Wiemken, Daniel F. Hoft, Christine Jacobs, John E. Morley.

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
