## [Decision Letter · Decision Letter 0]

24 Jun 2021

PONE-D-21-08521

Herpes Zoster Vaccination and Dementia

PLOS ONE

Dear Dr. Jeffrey Scherrer,

Thank you for submitting your manuscript to PLOS ONE. After careful consideration, we feel that it has merit but does not fully meet PLOS ONE’s publication criteria as it currently stands. Therefore, we invite you to submit a revised version of the manuscript that addresses the points raised during the review process.

We look forward to receiving your revised manuscript.

Kind regards,

Ping-Hsun Wu, M.D. PhD.

Academic Editor

PLOS ONE

Journal Requirements:

2.Please provide additional details regarding participant consent. In the ethics statement in the Methods and online submission information, please ensure that you have specified (1) whether consent was informed and (2) what type you obtained (for instance, written or verbal, and if verbal, how it was documented and witnessed). If your study included minors, state whether you obtained consent from parents or guardians. If the need for consent was waived by the ethics committee, please include this information.

3. Please modify the title to ensure that it is meeting PLOS’ guidelines (https://journals.plos.org/plosone/s/submission-guidelines#loc-title). In particular, the title should be "specific, descriptive, concise, and comprehensible to readers outside the field" and in this case we feel it is not informative and specific about your study's scope and methodology.

4.We note that you have indicated that data from this study are available upon request. PLOS only allows data to be available upon request if there are legal or ethical restrictions on sharing data publicly. For information on unacceptable data access restrictions, please see http://journals.plos.org/plosone/s/data-availability#loc-unacceptable-data-access-restrictions.

5.Thank you for stating the following in the Acknowledgments Section of your manuscript:

"Support for VA/CMS data is provided by the Department of Veterans Affairs, Veterans Health

362 Administration, Office of Research and Development, Health Services Research and

363 Development, VA Information Resource Center (Project Numbers SDR02-237 and 98-004). This

364 material is the result of work supported with resources and the use of facilities at the Harry S.

365 Truman Memorial Veterans' Hospital

366 The funder had no role in the design and conduct of the study; collection, management, analysis,

367 and interpretation of the data; preparation, review, or approval of the manuscript; and decision to

368 submit the manuscript for publication."

 "Benter Foundation award 2020-01 to JFS

https://benterfoundation.org/

The sponsor played no role in study design, data collection, analysis, decision to publish or preparation of the manuscript"

Additional Editor Comments:

Overall, this study is a well-designed study to evaluate the association between herpes zoster vaccination and dementia. The relationship between herpes zoster vaccination and dementia type (such as Alzheimer's disease versus non-Alzheimer's disease or vascular dementia) will be interesting. Some discussion can surround this issue.

Reviewers' comments:

Reviewer's Responses to Questions

**Comments to the Author**

1. Is the manuscript technically sound, and do the data support the conclusions?

Reviewer #1: Yes

Reviewer #2: Yes

2. Has the statistical analysis been performed appropriately and rigorously? 

Reviewer #1: Yes

Reviewer #2: Yes

3. Have the authors made all data underlying the findings in their manuscript fully available?

Reviewer #1: No

Reviewer #2: Yes

4. Is the manuscript presented in an intelligible fashion and written in standard English?

Reviewer #1: Yes

Reviewer #2: Yes

5. Review Comments to the Author

Reviewer #1: I have read the manuscript Herpes Zoster Vaccination and Dementia. Overall, this is an interesting and well performed study with a timely and important research question. Major strengths is the large sample size, use of two different cohorts and long follow-up time. The statistical analysis is sound. Major limitations are discussed and the conclusions are fair. I recommend this manuscrip is sent for revision.

My specific comments:

Previous studies on vaccinations and dementia all has major problems and should be discussed taking this into account. That having problems remembering things is associated with dementia development is not surprising, and therefore to rely on self-reported vaccinations is not without problems. Some other studies have very short follow-up time. As dementia disorders develop over several years before diagnosis, an effect seen in only one year for sure do not represent an actual causal effect but some confounding or reverese causality effect.

When referring to studies on infections and dementia, herpes simplex type one should be mentionned. Compared to Pertussis, there is very much more epidemiological support for a link to herpes simplex, see for example cohort studies by Letenneur 2008, Lovheim 2015, Lopatko-Lindman 2019, Linard 2020, and Itzhaki 2016 "Microbes and Alzheimer's Disease" for a review. I do not think Bordetella Pertussis deserves to be mentioned at all, it is only one of many pathogens suggested to be linked to AD development (other examples are chlamydia pneumonie, borrelia, toxoplasma, picorna virus and many others). Only HSV1 has been shown to associate with increased risk in several different cohort studies.

VZV could also be linked to vascular diseases and stroke. Maybe could this contribute to an effect on vascular dementa, and should possibly be discussed also?

Reviewer #2: The research design and analysis approach author took is very suitable, but there are some relatively some minor issuess that needs attention, therefore provided advice as follow:

(1)The title of the research is too simple, it is recommended to provide more information, for example the research design.

(2)Typesetting of the 282 line.

(3)The p-value location in Table 2 is strange, needs adjustment after verifying with the author.

(4)Please add hypothesis testing p-value of comparsion incidence rate in Table 3 and Table S.5.

(5)Supplementary Table3、Table4 Lacks the description for method of how to choose comparison group to do modeling; lacks the report of HZ infection original number of people, unsure of the result if HZ infection is weighted.

(6)There might have been a typographical error on supplementary Figure1b CY09~FY19 , needs adjustment after verifying with the author.

(7)Supplementary Figure1a Figure1b both specially mentioned that samples are divided into two groups of eligible or not eligible, but why not fails to further analysis?

6. PLOS authors have the option to publish the peer review history of their article (what does this mean?). If published, this will include your full peer review and any attached files.

Reviewer #1: No

Reviewer #2: No

---

## [Author Response · Author response to Decision Letter 0]

2 Jul 2021

RESPONSE TO CRITIQUES

Editor comments

1) Format manuscript to PLOS ONE style

We have formatted the manuscript to adhere to PLOS ONE style

2.Please provide additional details regarding participant consent. In the ethics statement in the Methods and online submission information, please ensure that you have specified (1) whether consent was informed and (2) what type you obtained (for instance, written or verbal, and if verbal, how it was documented and witnessed). If your study included minors, state whether you obtained consent from parents or guardians. If the need for consent was waived by the ethics committee, please include this information. If you are reporting a retrospective study of medical records or archived samples, please ensure that you have discussed whether all data were fully anonymized before you accessed them and/or whether the IRB or ethics committee waived the requirement for informed consent. If patients provided informed written consent to have data from their medical records used in research, please include this information.

We have added the following statement to the Methods and Ethics Statement: The Saint Louis University IRB approved the research as a non-human subjects study because this was a retrospective cohort study of anonymized medical record and claims data. Investigators only had access to anonymized data.

We report this statement in the submission form.

3. Please modify the title to ensure that it is meeting PLOS’ guidelines (https://journals.plos.org/plosone/s/submission-guidelines#loc-title). In particular, the title should be "specific, descriptive, concise, and comprehensible to readers outside the field" and in this case we feel it is not informative and specific about your study's scope and methodology.

We changed the title to: “Impact of Herpes Zoster Vaccination on Incident Dementia: A Retrospective Study in Two Patient Cohorts”

4.We note that you have indicated that data from this study are available upon request. PLOS only allows data to be available upon request if there are legal or ethical restrictions on sharing data publicly. For information on unacceptable data access restrictions, please see http://journals.plos.org/plosone/s/data-availability#loc-unacceptable-data-access-restrictions.

We now submit a Data Availability Statement: Veterans Health Administration (VHA) may not be shared without interested persons obtaining VHA IRB and Data Access Request approvals. MarketScan data is proprietary and the authors are prevented from sharing data per data use agreement.

5.Thank you for stating the following in the Acknowledgments Section of your manuscript:

"Support for VA/CMS data is provided by the Department of Veterans Affairs, Veterans Health

362 Administration, Office of Research and Development, Health Services Research and

363 Development, VA Information Resource Center (Project Numbers SDR02-237 and 98-004). This

364 material is the result of work supported with resources and the use of facilities at the Harry S.

365 Truman Memorial Veterans' Hospital

366 The funder had no role in the design and conduct of the study; collection, management, analysis,

367 and interpretation of the data; preparation, review, or approval of the manuscript; and decision to

368 submit the manuscript for publication."

"Benter Foundation award 2020-01 to JFS

https://benterfoundation.org/

The sponsor played no role in study design, data collection, analysis, decision to publish or preparation of the manuscript" 

We removed the funding statement from the Acknowledgements and we do not need to revise the funding statement in the online submission.

A list of captions for supporting information is now presented at the end of our manuscript.

7. Additional Editor Comments:

Overall, this study is a well-designed study to evaluate the association between herpes zoster vaccination and dementia. The relationship between herpes zoster vaccination and dementia type (such as Alzheimer's disease versus non-Alzheimer's disease or vascular dementia) will be interesting. Some discussion can surround this issue.

We expanded the Discussion, paragraph 1, with the following statements: Our sensitivity analysis indicate that the reduced risk for dementia also applies to AD. While the magnitude of effect was marginally less for risk of AD, our results support an association between HZ vaccination and reduced risk for dementia, including AD. Vaccinations may reduce risk for other forms of dementia, such as vascular dementia, by lowering inflammation which is common in cardiovascular conditions.

Reviewer #1: I have read the manuscript Herpes Zoster Vaccination and Dementia. Overall, this is an interesting and well performed study with a timely and important research question. Major strengths is the large sample size, use of two different cohorts and long follow-up time. The statistical analysis is sound. Major limitations are discussed and the conclusions are fair. I recommend this manuscript is sent for revision.

Thank you for the kind remarks.

1) Previous studies on vaccinations and dementia all has major problems and should be discussed taking this into account. That having problems remembering things is associated with dementia development is not surprising, and therefore to rely on self-reported vaccinations is not without problems. Some other studies have very short follow-up time. As dementia disorders develop over several years before diagnosis, an effect seen in only one year for sure do not represent an actual causal effect but some confounding or reverese causality effect.

We mention the limitations of existing research in the Introduction and now add additional comments about these studies in the Discussion, lines 326-328.

When referring to studies on infections and dementia, herpes simplex type one should be mentionned. Compared to Pertussis, there is very much more epidemiological support for a link to herpes simplex, see for example cohort studies by Letenneur 2008, Lovheim 2015, Lopatko-Lindman 2019, Linard 2020, and Itzhaki 2016 "Microbes and Alzheimer's Disease" for a review. I do not think Bordetella Pertussis deserves to be mentioned at all, it is only one of many pathogens suggested to be linked to AD development (other examples are chlamydia pneumonie, borrelia, toxoplasma, picorna virus and many others). Only HSV1 has been shown to associate with increased risk in several different cohort studies. VZV could also be linked to vascular diseases and stroke. Maybe could this contribute to an effect on vascular dementa, and should possibly be discussed also?

We revised this section of the Introduction which now reads: “Numerous bacterial and viral infections have been associated with incident dementia and or AD[2, 3]. Among these, the largest body of evidence supports a link between herpes zoster (HZ) infection and increased risk for dementia[5-10].”

Additional references suggested by the reviewer have been added.

Reviewer #2: The research design and analysis approach author took is very suitable, but there are some relatively some minor issuess that needs attention, therefore provided advice as follow:

(1)The title of the research is too simple, it is recommended to provide more information, for example the research design.

See response to Editor’s comment

(2)Typesetting of the 282 line.

I could not find a typesetting issue in line 282. But I did add an indent to line 283.

(3)The p-value location in Table 2 is strange, needs adjustment after verifying with the author.

There was one p-value that we moved into the correct row. Otherwise we are not sure what the reviewer saw in Table 2.

(4)Please add hypothesis testing p-value of comparison incidence rate in Table 3 

and Table S.5.

We have now added p-values for cumulative incidence and incidence rate in Tables 3 and S.5. 

(5)Supplementary Table 3 and Table 4 Lacks the description for method of how to choose comparison group to do modeling; lacks the report of HZ infection original number of people, unsure of the result if HZ infection is weighted.

We have tried to add more description to these tables (adding rows to describe models – age stratified vs. race stratified, adding HR column headers). We have added a footnote indicating that the hazard ratio within each group of patients identified is comparing HZ vaccination vs. no vaccination. The results in these tables include crude and various weighted models as indicated in the column headers. These tables present information included in the forest plot Figure 1. 

(6)There might have been a typographical error on supplementary Figure1b CY09~FY19 , needs adjustment after verifying with the author.

Thank you for catching this error. We have corrected it. 

(7)Supplementary Figure1a Figure1b both specially mentioned that samples are divided into two groups of eligible or not eligible, but why not fails to further analysis?

The sampling scheme used 2 index dates to identify patients, either CY2011 (FY2011 in VHA) or CY2012 (FY2012 in VHA). Patients that were not eligible on the first index date (as indicated by the boxes “Sample that is not eligible 10/1/2010” (VHA) or “Sample that is not eligible 1/1/2011” (MarketScan)) were assessed for eligibility at the next index date. Also for clarity, on page 7 lines 119-120 of the manuscript, we have mentioned that “Patients not meeting eligibility criteria in 2011 were used to sample for the 2012 index date”. We have added footnotes to both figures with similar verbiage to add clarity to these figures. This sampling scheme tries to show that a patient entered the cohort the first time he or she met eligibility criteria (either at the first or second index date).

---

## [Decision Letter · Decision Letter 1]

24 Aug 2021

PONE-D-21-08521R1

Impact of Herpes Zoster Vaccination on Incident Dementia: A Retrospective Study in Two Patient Cohorts

PLOS ONE

Dear Dr. Jeffrey Scherrer,

Thank you for submitting your manuscript to PLOS ONE. After careful consideration, we feel that it has merit but does not fully meet PLOS ONE’s publication criteria as it currently stands. Therefore, we invite you to submit a revised version of the manuscript that addresses the points raised during the review process.

We look forward to receiving your revised manuscript.

Kind regards,

Ping-Hsun Wu, M.D. PhD.

Academic Editor

PLOS ONE

Journal Requirements:

Additional Editor Comments (if provided):

According to reviewers' suggestion, some minor revision is considered before acceptance of this manuscript. Some studies regarding the link between VZV infection and AD risk could be included and discussed. Besides, please check and revised the p-value position in Table 2. The p-value should be presented in Table 3 according to reviewer 2's suggestion.

Reviewers' comments:

Reviewer's Responses to Questions

**Comments to the Author**

1. If the authors have adequately addressed your comments raised in a previous round of review and you feel that this manuscript is now acceptable for publication, you may indicate that here to bypass the “Comments to the Author” section, enter your conflict of interest statement in the “Confidential to Editor” section, and submit your "Accept" recommendation.

Reviewer #1: (No Response)

Reviewer #3: (No Response)

2. Is the manuscript technically sound, and do the data support the conclusions?

Reviewer #1: Yes

Reviewer #3: Yes

3. Has the statistical analysis been performed appropriately and rigorously? 

Reviewer #1: Yes

Reviewer #3: N/A

4. Have the authors made all data underlying the findings in their manuscript fully available?

Reviewer #1: No

Reviewer #3: Yes

5. Is the manuscript presented in an intelligible fashion and written in standard English?

Reviewer #1: Yes

Reviewer #3: Yes

6. Review Comments to the Author

Reviewer #1: I have once again reviewed this manuscript, and overall I think the manuscript is now close to being acceptable for publication. I hav a few suggestions still, outlined below, but if these are properly taken into account I think the manuscript can then be accepted. The material is large, the study of high quality and the research question important.

At line 72, the authors have confused herpes zoster (VZV) with herpes simplex (HSV1 and HSV2). The references, and many other studies supporting a link to herpes virus, are all to herpes simplex. This sentence therefore must be corrected, and zoster changed to simplex. The reference Lopatko Lindman 2019 (Lopatko Lindman K, et al. Alzheimers Dement (N Y). 2019 Nov 4;5:697-704. doi: 10.1016/j.trci.2019.09.014. eCollection 2019. PMID: 31921962 ) should be included as this study also indicate a strong interaction between HSV1 and APOE4.

Recent registry-based studies indicating a link between VZV infection and AD risk (and effect of antivirals) should possibly also be included (Bae et al, Lopatko-Lindman et al, Chen et al), as these do show a possible link to VZV infections in particular.

The short follow-up time of previous registry-based studies on vaccinations and dementia (influenza vaccination) should be mentioned and problematized.

The apparent very small effect of the vaccinations on the risk of later VZV infection should possibly be discussed. Are these vaccinations effective enough?? Still, even if the risk is not decreased much, infections might be less severe, and complication risk might still decrease.

Reviewer #3: The author has made a few adjustment correspond to the advice from reviewer #2, but there were some minor issues that weren’t answered:

“(3)The p-value location in Table 2 is strange, needs adjustment after verifying with the author.

There was one p-value that we moved into the correct row. Otherwise we are not sure what the reviewer saw in Table 2.”

I would like the author to address the reason why the p-value are stated in subset instead of their correspond categorical variable, e.g., in the Region the p-value is stated within the south region.

“4)Please add hypothesis testing p-value of comparison incidence rate in Table 3

and Table S.5.

We have now added p-values for cumulative incidence and incidence rate in Tables 3 and S.5.”

I did not see p-value of incidence rate comparison between No vaccination group and HZ vaccination group added in Table 3 as stated above, there are 8 comparisons, therefore 8 p-values needs to be added in table 3.

7. PLOS authors have the option to publish the peer review history of their article (what does this mean?). If published, this will include your full peer review and any attached files.

Reviewer #1: No

Reviewer #3: No

---

## [Author Response · Author response to Decision Letter 1]

25 Aug 2021

RESPONSE TO CRITIQUES

Editor comments

Additional Editor Comments (if provided):

According to reviewers' suggestion, some minor revision is considered before acceptance of this manuscript. Some studies regarding the link between VZV infection and AD risk could be included and discussed. Besides, please check and revised the p-value position in Table 2. The p-value should be presented in Table 3 according to reviewer 2's suggestion.

We responded positively to these limitations. Detailed responses follow.

Track changes were made in the main document. I highlighted the p-values which were already in Table S.5. and I provided a non-highlighted supplemental file.

Reviewer #1: 

I have once again reviewed this manuscript, and overall I think the manuscript is now close to being acceptable for publication. I have a few suggestions still, outlined below, but if these are properly taken into account I think the manuscript can then be accepted. The material is large, the study of high quality and the research question important.

1) At line 72, the authors have confused herpes zoster (VZV) with herpes simplex (HSV1 and HSV2). The references, and many other studies supporting a link to herpes virus, are all to herpes simplex. This sentence therefore must be corrected, and zoster changed to simplex. The reference Lopatko Lindman 2019 (Lopatko Lindman K, et al. Alzheimers Dement (N Y). 2019 Nov 4;5:697-704. doi: 10.1016/j.trci.2019.09.014. eCollection 2019. PMID: 31921962 ) should be included as this study also indicate a strong interaction between HSV1 and APOE4.

Recent registry-based studies indicating a link between VZV infection and AD risk (and effect of antivirals) should possibly also be included (Bae et al, Lopatko-Lindman et al, Chen et al), as these do show a possible link to VZV infections in particular.

The reviewer refers to the following sentence: “Among these, the largest body of evidence supports a link between herpes infection, and increased risk for dementia[5-10].”

We revised this section to be more specific. It now reads: Herpes simplex virus has been associated with increased risk for dementias (Lopatko-Lindman et al. 2020, Lovheim et al. 2015, Honjo et al. 2009). In addition, genetic risk markers for Alzheimer’s Disease (AD) have been shown to interact with herpes simplex virus to increase likelihood of developing AD (Lopatko-Lindman 2019). In large health insurance databases and patient registries, herpes zoster (HZ) infection also has been associated with an increased risk for dementia (Chen et la. 2018, Bae et al 2020,); those with vs. without herpes zoster ophthalmicus had nearly a 3-fold increased risk for developing dementia (Tsai et al. 2017). Furthermore, patients with a history of herpes simplex or HZ infection and received antivirals, have a lower risk for dementia than patients with a history of these infections who did not receive antivirals (Lopatko-Lindman et al. 2020, Chen et al. 2018, Bae et al. 2020, ). 

2) The short follow-up time of previous registry-based studies on vaccinations and dementia (influenza vaccination) should be mentioned and problematized.

The two influenza vaccination studies from the Taiwan national health insurance databases Luo et al. and Liu et al. had different follow-up times. Liu et al. 2016 used a 12 month follow-up equal to >17,000 person years in unvaccinated and >35,000 in vaccinated patients. The second study by Luo et al used the same data registry and followed patients for up to 12 years. The adjusted HR for the association between any influenza vaccination and risk for dementia was 0.68 in both studies. We believe the follow-up time did not bias results in these studies, but overall, we agree that shorter follow-up times may under-estimate the association between vaccination and dementia when time to develop the outcome is insufficient.

3) The apparent very small effect of the vaccinations on the risk of later VZV infection should possibly be discussed. Are these vaccinations effective enough?? Still, even if the risk is not decreased much, infections might be less severe, and complication risk might still decrease.

We have added the following text to the Discussion: “We observed that in VHA patients, 7.4% of patients without HZ vaccination and 6.9% of patients with HZ vaccine had at least one ICD diagnostic code for HZ infection in follow-up. This suggests a modest effect of these vaccinations on preventing HZ. This finding, that vaccination had little impact on HZ infection in our studied populations may be explained by the possibility that preventing HZ infection/reactivation is not the mechanism by which vaccination is associated with lower dementia risk.”

4) The author has made a few adjustment correspond to the advice from reviewer #2, but there were some minor issues that weren’t answered:

-The p-value location in Table 2 is strange, needs adjustment after verifying with the author.

-There was one p-value that we moved into the correct row. Otherwise we are not sure what the reviewer saw in Table 2.”

-I would like the author to address the reason why the p-value are stated in subset instead of their correspond categorical variable, e.g., in the Region the p-value is stated within the south region.

The p-values were based on an omnibus test (example region*vaccination), not on a subset. We apologize for causing confusion due to placement of p-values. P-values for categorical variables are now placed in appropriate rows in Table 2.

5) Please add hypothesis testing p-value of comparison incidence rate in Table 3

and Table S.5.

We have now added p-values for cumulative incidence and incidence rate in Tables 3 and S.5.”

6) I did not see p-value of incidence rate comparison between No vaccination group and HZ vaccination group added in Table 3 as stated above, there are 8 comparisons, therefore 8 p-values needs to be added in table 3.

The p-values for all cumulative incidence and incidence rate comparisons are now included in Table 3

---

## [Decision Letter · Decision Letter 2]

1 Sep 2021

Impact of Herpes Zoster Vaccination on Incident Dementia: A Retrospective Study in Two Patient Cohorts

PONE-D-21-08521R2

Dear Dr. Jeffrey Scherrer,

We’re pleased to inform you that your manuscript has been judged scientifically suitable for publication and will be formally accepted for publication once it meets all outstanding technical requirements.

Kind regards,

Ping-Hsun Wu, M.D. PhD.

Academic Editor

PLOS ONE

Additional Editor Comments (optional):

All suggestions had been revised accordingly. This manuscript is available for publication.

Reviewers' comments:

Reviewer's Responses to Questions

**Comments to the Author**

1. If the authors have adequately addressed your comments raised in a previous round of review and you feel that this manuscript is now acceptable for publication, you may indicate that here to bypass the “Comments to the Author” section, enter your conflict of interest statement in the “Confidential to Editor” section, and submit your "Accept" recommendation.

Reviewer #1: All comments have been addressed

Reviewer #3: All comments have been addressed

2. Is the manuscript technically sound, and do the data support the conclusions?

Reviewer #1: Yes

Reviewer #3: Yes

3. Has the statistical analysis been performed appropriately and rigorously? 

Reviewer #1: Yes

Reviewer #3: Yes

4. Have the authors made all data underlying the findings in their manuscript fully available?

Reviewer #1: No

Reviewer #3: Yes

5. Is the manuscript presented in an intelligible fashion and written in standard English?

Reviewer #1: Yes

Reviewer #3: Yes

6. Review Comments to the Author

Reviewer #1: (No Response)

Reviewer #3: (No Response)

7. PLOS authors have the option to publish the peer review history of their article (what does this mean?). If published, this will include your full peer review and any attached files.

Reviewer #1: No

Reviewer #3: No

---

## [Editor Report · Acceptance letter]

5 Nov 2021

PONE-D-21-08521R2 

Impact of Herpes Zoster Vaccination on Incident Dementia: A Retrospective Study in Two Patient Cohorts 

Dear Dr. Scherrer:

I'm pleased to inform you that your manuscript has been deemed suitable for publication in PLOS ONE. Congratulations! Your manuscript is now with our production department. 

Kind regards, 

on behalf of

Dr. Ping-Hsun Wu 

Academic Editor

PLOS ONE